# Burrowing Behavior as Robust Parameter for Early Humane Endpoint Determination in Murine Models for Pancreatic Cancer

**DOI:** 10.3390/ani15091241

**Published:** 2025-04-28

**Authors:** Jakob Brandstetter, Lisa Hoffmann, Ingo Koopmann, Tim Schreiber, Benjamin Schulz, Stephan Patrick Rosshart, Dietmar Zechner, Brigitte Vollmar, Simone Kumstel

**Affiliations:** 1Rudolf-Zenker-Institute of Experimental Surgery, University Medical Center Rostock, 18057 Rostock, Germany; jakob.brandstetter@uni-rostock.de (J.B.); lisa.hoffmann@uni-rostock.de (L.H.); ingo.koopmann@uni-rostock.de (I.K.); tim.schreiber@med.uni-rostock.de (T.S.); dietmar.zechner@uni-rostock.de (D.Z.); brigitte.vollmar@med.uni-rostock.de (B.V.); 2Department of Microbiome Research, University Hospital Erlangen, Friedrich-Alexander-Universität Erlangen-Nürnberg (FAU), 91054 Erlangen, Germany; stephan.rosshart@uk-erlangen.de; 3Department of Medicine 1, University Hospital Erlangen, Friedrich-Alexander-Universität Erlangen-Nürnberg (FAU), 91054 Erlangen, Germany

**Keywords:** animal welfare, refinement, early humane endpoint, pancreatic cancer

## Abstract

In vivo experiments remain essential for testing new therapeutic approaches against fatal diseases, such as pancreatic cancer. Laws and regulations mandate the continuous improvement of the animal’s welfare during these experiments. One important aspect is the definition of early humane endpoint criteria to prevent the severe suffering of mice through timely euthanasia. In the present study, the welfare of mice in different pancreatic cancer models, and after various treatments for testing their preclinical efficacy, was monitored using body weight change, distress score, perianal temperature, burrowing behavior, nesting activity, and the mouse grimace scale. The ability of each parameter to predict the humane endpoint for each mouse was retrospectively quantified using receiver operating characteristic curve analysis. Burrowing behavior proved to be a robust predictor of the humane endpoint across all different pancreatic cancer models and treatment groups.

## 1. Introduction

Pancreatic ductal adenocarcinoma (PDA) ranks among the cancer entities with the lowest 5-year survival rates, at approximately 12%, due to late diagnoses and limited treatment options [1]. Projections for cancer-caused death see pancreatic cancer in second place by 2030 [2]. Patients with the option for surgical resection have the best chances for long-term survival [3]. However, PDA is typically diagnosed at an advanced, unresectable stage of tumor progression [4,5]. Therefore, alternative therapeutic approaches are urgently required to enhance the overall survival of the vast majority of patients. After passing initial in vitro tests, new therapeutic regimens need to be tested in vivo [6].

Animal experiments are mandatory for research on diseases and the development of effective therapies [7]. The overwhelming majority of preclinical studies are based on mice, with various models having emerged for studying pancreatic cancer [8,9,10]. In cancer research, cell line-based animal models account for the largest share (80%), due to their low cost and good reproducibility [8]. By application of “wildlings”, Rosshart and colleagues combined the advantages of a standardized genotype with a natural microbiome and pathogens [11]. The wildlings phenocopy the humane immune response better than common laboratory strains and might represent an important tool for increasing the translatability of preclinical data to humans [12].

Overall survival and progression-free survival are the primary endpoints from clinical cancer trials [13,14], and surrogate endpoints, such as response rate or time to progression, are unable to predict survival [15]. To allow the translatability, many preclinical cancer studies still rely on survival as the primary endpoint [16,17]. However, during these experiments, animals will be euthanized when they reach humane endpoints, clearly pre-defined by various criteria, e.g., losing > 20% body weight, hunched posture, decreased activity, or hypothermia [18,19,20]. These humane endpoints are mostly associated with a significant burden for animals, caused by tumor progression or side effects of applied therapeutic interventions. Furthermore, pain influences physiological processes and represents a potential bias in the scientific results [21,22].

The establishment of early humane endpoints, which can predict the occurrence of the originally pre-defined endpoint criteria a few days in advance, could contribute to a significant improvement in animal welfare in these studies [23,24]. Hence, the use of early humane endpoints serves the refinement aspect of the 3R-principles by preventing severe distress in laboratory animals [25]. Since it is in the nature of mice to suppress any signs of pain and discomfort, many non-invasive physical and behavioral parameters need to be tested for their predictive ability to determine humane endpoints. In addition to the assessment of body weight and the clinical score, the monitoring of perianal temperature, the mouse grimace scale (MGS), as well as burrowing behavior and nesting activity proved to be sensitive indicators for recognizing distress in many mouse models [26,27,28].

In the present study, we analyzed animal welfare using the above-mentioned parameters in a variety of cell-line-based pancreatic cancer models, utilizing different mouse strains, wildlings, and using different therapeutic interventions to quantify their effect in a preclinical setting. The predictive power of each parameter for humane endpoint determination was retrospectively calculated by using receiver operating characteristic curve analysis. This study aimed to identify a robust parameter for early humane endpoint prediction.

## 2. Materials and Methods

### 2.1. Cell Lines and Cell Culture

The murine pancreatic cancer cell line 6606PDA (gift from Prof. Tuveson, University of Cambridge, UK [29]) was cultured in Dulbecco’s Modified Eagle’s Medium (DMEM, 4.5 g/L Glucose, Biochrom GmbH, Berlin, Germany), supplemented with 10% fetal calf serum (FCS) and 1% Penicillin–Streptomycin (PAN-Biotech GmbH, Aidenbach, Germany). The murine pancreatic cancer cells, Panc02, were provided by the National Cancer Institute and cultured in 1640-RPMI Medium (Sigma-Aldrich, St. Louis, MO, USA), 10% FCS, and 1% Penicillin–Streptomycin (PAN-Biotech GmbH, Aidenbach, Germany). Both cell lines were incubated at 37 °C and 5% CO_2_. Before tumor cell injection, the cells were suspended in either PBS or a PBS-matrigel^®^-mix (354248, BD, Franklin Lakes, NJ, USA).

### 2.2. Animals

All animal experiments were approved by the local authority (Landesamt für Landwirtschaft, Lebensmittelsicherheit und Fischerei Mecklenburg-Vorpommern, 7221.3-1-16/21-4 (12 April 2021), 7221.3-1-10/21 (12 August 2021). The C57BL/6J and C57BL/6NTac mice were originally purchased from Charles River Laboratories (Wilmington, MA, USA) and bred in our animal facility under specific pathogen-free conditions. Health monitoring of these mice was routinely performed, according to FELASA Guidelines. In the last two years, *Helicobacter* sp., *Rodentibacter pneumotropicus*, and *Murine norovirus* were detected in some mice, and these mice were not used for breeding or experiments. Breeding pairs of C57BL/6NTac/wild mice (wildlings) were provided by Prof. Rosshart (University Hospital Erlangen, Erlangen, Germany). The wildlings were generated as previously described [11,12]. In short, embryos of C57BL/6NTac mice were generated in vivo, isolated, stored, and further transferred to pseudo-pregnant female wild mice. The genetics of wildlings are homogenous to C57BL/6NTac mice. However, their microbiome and their level of pathogen exposure resemble wild mice and can be kept stable over many generations [11,12]. These mice indicate a closer immune response to humans, compared to pathogen-poor laboratory mice, and therefore increase the translatability of preclinical data [12]. The wildlings were further bred in our facility. The breeding of and experiments with wildlings were performed in separate rooms and laboratories to avoid contamination from other laboratory mice. The wildlings were bred in type II cages (Zoonlab GmbH, Castrop-Rauxel, Germany) with filter tops. After weaning of the mice at 4–5 weeks, the pups were separated by gender and kept in bigger groups in type IV cages (Zoonlab GmbH, Castrop-Rauxel, Germany). To enable stable immunization, non-autoclaved hay and two tablespoons of compost were placed in the home cage weekly, during the age of 5–10 weeks. Afterwards, the wildlings were kept under the same conditions compared to the other laboratory mice. In the rooms of the wildlings, the following pathogens were detected within the last two years: *Helicobacter ganmani*, *Helicobacter mastomyrinus*, *Helicobacter genus*, *R. heylii*, *R. pneumoniae*, *Mite*, and *Pinworm*.

During the experiments, all the mice were kept and single-housed in type III cages (Zoonlab GmbH, Castrop-Rauxel, Germany) with filter tops, in a 12-h dark–light cycle, with a room temperature of 21 ± 2 °C, a relative humidity of 60 ± 20%, food (pellets, 10 mm, ssniff-Spezialdiäten GmbH, Soest, Germany), and tap water, with or without analgesic *ad libitum*. Enrichment was provided by supplying nesting material (shredded tissue paper, Verbandmittel GmbH, Frankenberg, Germany) and paper rolls (75 × 38 mm, H 0528–151, ssniff-Spezialdiäten GmbH), as well as wooden sticks (40 × 16 × 10 mm, Abedd, Vienna, Austria).

### 2.3. Pancreatic Cancer Models and Therapeutic Strategies

Before tumor cell injection, each mouse was initially anesthetized with 3–5 vol.% isoflurane in a box. The anesthesia was continued over a mask with 1–2.5 vol.% isoflurane. All mice were positioned on a heating plate at 37 °C during the procedure, and their eyes were kept wet by eye ointment. As an analgesic, 3 mg/mL metamizole was added to the drinking water of all the mice at least one week before tumor cell injection and continuously administered until the end of the experiments.

The subcutaneous pancreatic cancer model was induced, as described before [26], by subcutaneous injection of 5 × 10^5^ Panc02 cells in 100 µL PBS into the shaved right and left flanks of each mouse. Four male and five female C57BL/6J mice were used for the subcutaneous model, with an age of 16–20 weeks.

For the intravenous model, 1 × 10^6^ Panc02 cells in 50 µL PBS were injected into the tail vein using a 30G needle (BD, Franklin Lakes, NJ, USA) and a catheter (ICU Medical, Inc., San Clemente, CA, USA). In this model, five male and four female C57BL/6J mice with an average age of 18–21 weeks were used.

For the orthotopic tumor cell injection of the Panc02 cells without any subsequent therapeutic intervention, four male and five female C57BL/6J mice (age: 20–21 weeks) were applied (Appendix A). Carprofen (5 mg/kg; Rimadyl, Pfizer GmbH, Berlin, Germany) was injected subcutaneously directly before surgery, as additional pain medication. The abdomen of the mice was shaved, disinfected, and the abdominal cavity was opened by a laparotomy. A total of 5 µL of the Panc02 cell suspension (1 × 10^4^ cells in 1:1 PBS and matrigel^®^) was injected with a 25 µL syringe (Hamilton, Reno, NE, USA) into the pancreas. The peritoneum was closed with a continuous suture, using a 5-0 Vicryl suture (Johnson & Johnson Medical GmbH, New Brunswick, NJ, USA). The epidermis was closed with separated surgical knots with 5-0 Prolene suture (Johnson & Johnson Medical GmbH). Some results from the above-mentioned experiments were already published in a different scientific context [26].

The orthotopic injection of Panc02 cells for the therapeutic treatment with erlotinib and LHX-254 was conducted in the same way as mentioned above. One week after orthotopic tumor cell injection, the mice were treated with either C-Raf inhibitor LHX-254 (35 mg/kg, MedChemExpress, Monmouth Junction, NJ, USA), EFGR inhibitor erlotinib (75 mg/kg, MedChemExpress, Monmouth Junction, NJ, USA), the combination of both therapeutics, or the vehicle five times a week by oral gavage, receiving a maximum of 30 doses. The compounds were dissolved in 90% PEG400 (Merck KGaA, Darmstadt, Germany) and 10% Tween80 (Sigma-Aldrich, St. Louis, USA). The C57BL/6J mice were between 12 and 26 weeks old. The mice were randomized into the treatment groups with three male and two female mice in the vehicle group, four female and three male mice receiving erlotinib, three male and three female mice assigned to the LXH-254 group, and two female and four male mice in the group with combination therapy (Appendix A). According to the previously published score sheet, the mice were euthanized when reaching humane endpoint criteria [26].

The 6606PDA cells were injected orthotopically into the pancreas and intravenously into the tail vein. Both injections were performed as described above. A total of 5 µL of 2.5 × 10^5^ 6606PDA cells in PBS:Matrigel^®^ (1:1) were injected into the pancreas, and 50 µL cell suspension (7 × 10^6^ cells/mL) in PBS were injected into the tail vein of either male C57BL/6NTac (10–18 weeks old) or wildlings (19–22 weeks old). On the fourth day after surgery, these mice were randomized into the treatment groups and treated either with the vehicle solution, the dual combination of *Kras*-inhibitor BI-3406 and MEK-inhibitor trametinib, or the triple combination, adding pan-*PI3K* inhibiting agent BKM120. These therapeutics were dissolved in the vehicle solution containing 60% Phosal50PG (Lipoid GmbH, Ludwigshafen, Germany), 30% PEG400 (Merck KGA, Darmstadt, Germany), and 10% ethanol. The therapeutics were administered via oral gavage five times a week until day 36 after tumor cell injection. Ten C57BL/6NTAc mice were treated with the vehicle, eight mice received BI-3406 and trametinib, and nine mice received BI-3406, trametenib, and BKM-120. However, in the present study, only data from the non-survivors were used for the determination of early humane endpoint criteria. These mice had to be euthanized due to the occurrence of pre-defined humane endpoint criteria (e.g., 20% body weight loss or distinct apathy [30]) during days 18–36 of tumor cell progression. The non-survivors included six mice from the vehicle group, two mice from the dual therapy, and three mice from the triple treatment (Appendix A).

Analogous to the therapy experiment with the C57BL/6NTAc mice, the same experiment was carried out with male wildlings. Originally, ten wildlings were treated with the vehicle, nine mice received the combination therapy of trametinib and BI-3406, and ten wildlings were treated additionally with BKM-120. For the present study, data from the mice were used from the non-survivors, which had to be euthanized between days 18 and 36 during tumor progression due to the occurrence of humane endpoint criteria. Three wildlings were included as non-survivors in the vehicle group, and one mouse, respectively, for the dual and triple therapies (Appendix A).

### 2.4. Welfare Parameters

The body weight of mice was assessed daily before tumor cell injection (pre) and during the entire experiment until euthanasia. Likewise, the distress score was quantified daily, according to the score sheet. Two different score sheets were used for the present study. The mice injected with Panc02 cells were monitored by the recently published score sheet (Appendix A) [26]. For the experiments with the metastasizing PDA model, using the 6606PDA cells, an older score sheet was applied (Appendix A) [30]. Both score sheets include similar criteria, except the endpoint for body weight loss, where the new score sheet sets the endpoint from 20% to 15% loss from the initial body weight.

The perianal temperature was measured on healthy mice before tumor cell inoculation (pre), after tumor cell injection, on different time points during tumor progression (tp), and on the days before humane endpoint. The mice were placed on the cage lid, and a contactless infrared thermometer (WEPA Apothekenbedarf GmbH & Co. KG, Hillscheid, Germany) measured the perianal surface temperature three times [31]. The average of the three values was used for further data analysis.

Burrowing behavior was quantified by providing a burrowing tube (6.5 × 6.5 × 15.00 cm) in the home cage, filled with 200 g (±1 g) pellets (ssniff-Spezialdiäten GmbH, Soest, Germany) two to three hours before the dark phase. The burrowed amount of pellets was measured after 2 h and after 17 h the next morning [32]. Animals, which burrowed less than 50 g in 2 h or overnight before tumor cell injection, were excluded from the data analysis. A total of 11 out of 67 mice had to be excluded from the 2 h burrowing analysis (16%), and 5 out of 67 mice (7%) were excluded from the burrowing analysis overnight. The nesting activity was assessed with a cotton nestlet (5 cm square, ZOONLAB GmbH, Castrop-Rauxel, Germany), which was placed in the home cage 1–2 h before the dark phase. The nest was scored on the next morning, according to the 1–5 point scale from Deacon et al. [33]. Additionally, a score of 6 was given for a perfect nest, where 90% of the circumference of the nest wall is higher than the body height of the mouse. All healthy mice that created a nest and scored less than 4 before tumor cell injection were excluded from the analysis. Of the 67 mice, in total, 10 mice (15%) were excluded from data analysis due to poor nest-building behavior.

The mouse grimace scale (MGS) was quantified by placing the mice into a transparent polycarbonate box (9 × 5 × 5 cm). The box was placed into an illuminated tent and was lit from the front. The mice had 5 min to acclimatize in the box and were filmed for an additional 5 min with a digital single-lens reflex camera (Canon EOS 70D, Tokyo, Japan). Three pictures were captured from each video, randomized, and scored by three researchers in a blinded fashion, according to the score from Langfort et al. [34]. During the days before the humane endpoint, the mice were scored by three researchers in the home cage to avoid unnecessary stress for the animals. The scores were averaged, and the baseline values were subtracted for data analysis.

The welfare parameters, including the perianal temperature, burrowing behavior, nesting activity, and mouse grimace scale, were assessed before tumor cell injection (pre), after tumor cell injection (op, acute post op), and at different time points during tumor progression (tp). The days of analysis during the tumor progression differed between the distinct pancreatic cancer models, due to varying degrees of rapid tumor progression (Appendix A). The data on the distress analysis after tumor cell injection were not included in the present study, since the determination of early humane endpoints and the comparison of distress during the tumor progression is the focus of the present study. If mice had to be euthanized before assessing welfare parameters in the late phase, early humane endpoint values were compared to values from a middle phase assessment (Appendix A).

As soon as criteria from the score sheets were noticed (e.g., ruffled fur, abnormal posture, or 10% body weight loss), all parameters were assessed daily until the occurrence of the endpoint criteria on the respective score sheets. Additionally, the daily welfare assessment started as soon as 5% body weight loss was quantified within 24 h.

### 2.5. Statistics

Data for all Figures were analyzed and graphed with the program GraphPad Prism 8.4.3 (GraphPad Software, San Diego, CA, USA). Data for Figures 1, 3, 5, and 7 are presented as whisker plots indicating median, quartiles (25, 75), minimum and maximum, as well as single values. The data for each parameter were quantified for the mice in the specific groups before tumor cell injection (pre), during tumor progression (tp), and on the days before each individual humane endpoint. The statistical analyses were performed either with two-way ANOVA or a mixed model, followed by Dunnett’s test for comparison of each time point with the tumor progression (tp) values or Tukey’s test for inter-model differences. For some parameters, the statistical analysis was not possible due to missing values in certain groups at specific time points. Details of the statistical analysis are listed in Appendix A. To quantify the predictive power of each parameter for early humane endpoint determination, receiver operating characteristic (ROC) curve analysis was performed. For the ROC analysis, the values of each parameter quantified on the respective days before the humane endpoint were directly compared with values during tumor progression. The goal was to indicate the diagnostic ability of each parameter to differentiate between a sick mouse during tumor progression and a dying mouse before reaching the humane endpoint. The predictive power of each parameter is indicated via the AUC values in the heat maps for each murine PDA model or specific treatment groups in Figures 2, 4, 6, and 8. An AUC value of 0.5 indicates low predictive ability, and an AUC of 1.0 indicates the highest discriminatory power for early humane endpoint determination.

## 3. Results

The quantification of body weight change in the different Panc02-models revealed a significantly lower body weight in the orthotopically injected mice compared to the tumor progression phase, three days before the humane endpoint. The body weight change was significantly lower compared to the subcutaneously and intravenously injected mice three days before the humane endpoint and compared to the subcutaneous model on the day of the humane endpoint (Figure 1A). A slight increase in the distress score was observed two days before each individual humane endpoint. A significantly higher distress score was noticed for the orthotopic and intravenous animal models compared to the subcutaneously injected mice on the day of the humane endpoint (Figure 1B). The perianal surface temperature remained stable for all the different Panc02 cell-injected models on the days prior to the humane endpoint. At the individual humane endpoint for each mouse, a slight, non-significant drop in perianal temperature was noticed in some mice, especially after orthotopic and intravenous tumor cell injection (Figure 1C). The burrowing behavior after 2 and 17 h remained quite stable during tumor progression, followed by a strong drop three days prior to the humane endpoint in the different animal models (Figure 1D,E). A strong reduction in nesting activity was also quantified, especially in the orthotopically injected mice three days prior to the individual humane endpoint (Figure 1F). A non-significant increase in the MGS score was observed for all animal models at the individual humane endpoint (Figure 1G).

To analyze which welfare parameters were particularly appropriate to quantify early humane endpoints in the distinct models, receiver operating characteristic (ROC) curves were utilized. Therefore, all parameters assessed on the days prior to the humane endpoint and the day of the humane endpoint were compared with the values during tumor progression. Subsequently, false-positive rates and true-positive rates were displayed on the x-axis and the y-axis, respectively, and the sensitivity of each parameter was calculated via the area under the curve (AUC). The AUC values were displayed via heat maps, indicating the parameters of high predictive power in green (AUC: 0.80–1.00) and low predictive power (AUC: 0.50–0.70) in red. The heat maps were graphed separately for the orthotopic (Figure 2A), intravenous (Figure 2B), and subcutaneous animal models (Figure 2C). The body weight indicated a low discriminatory power up to two days prior to the humane endpoint in the three animal models. An increase in predictability for body weight could be observed one day before and on the humane endpoint, with an AUC of 0.83–0.96 when comparing all the animal models (Figure 2A–C). Similar results were obtained for the distress score, indicating mostly a low predictive power (AUC: 0.50–0.61) two days prior to the humane endpoint. An increase in predictability was noticed for the distress score one day prior to the humane endpoint in the orthotopic and intravenous models (AUC: 0.78–0.86). The perianal temperature started to be predictive two days before the humane endpoint in the intravenous model (AUC: 0.90–1.00), but one day prior to and on the humane endpoint, the diagnostic ability was rather low in the orthotopic and subcutaneous models (AUC: 0.52–0.72). The burrowing activity after 2 and 17 h indicated a high predictive power 3 days before the humane endpoint (AUC: 1.00) in the orthotopic model and 1–2 days prior to the humane endpoint in the intravenous and subcutaneous models. Nesting activity seemed to have a good endpoint prediction as well, with an AUC between 0.81–1.00 in the murine pancreatic cancer models 1–2 days prior to the humane endpoint. The discriminatory power of the MGS was distinctly different between the murine models (AUC: 0.50–1.00, Figure 2A–C).

For the C57BL/6J mice orthotopically injected with Panc02 cells and assigned to the different therapy groups, vehicle, LXH-254, erlotinib, or the combinatorial treatment, a slight reduction in body weight was noticed towards the humane endpoint (Figure 3A). No significant difference was observed between the different treatment groups. However, significantly reduced bodyweight, compared to the tumor progression, was quantified after erlotinib intervention and the combinatorial treatment, in addition to LXH-254, on the day of the humane endpoint (Figure 3A). A continuous, but non-significant increase in the distress score, compared to the tumor progression, was noticed for all the treatments in the orthotopic model (Figure 3B). A minimal reduction in perianal temperature could be noticed only on the day of the humane endpoint for the treatment groups (Figure 3C). A strong reduction in burrowing behavior after 2 and after 17 h was noticed at least three days prior to the humane endpoint in all groups (Figure 3D,E). The animals’ nesting activity decreased in all groups on the last two days, however, some mice displayed a high nesting score even on the humane endpoint itself (Figure 3F). A steady increase in the MGS score was quantified until the humane endpoint for all the treatment groups. Statistical analysis was carried out only for Figure 3A-B, as statistical analysis was not possible for the other parameters due to missing values (Figure 3D–G).

The ROC curve analysis revealed that body weight change indicates a good discriminatory power to differentiate between mice during tumor progression and mice indicating distress starting three days prior to the humane endpoint, especially for the vehicle, LXH-254, and combinatorial treatment groups (AUC: 0.81–1.00, Figure 4A–D). The predictive power of body weight proved to be rather low for humane endpoint determination of erlotinib-treated mice, even on the day of the humane endpoint (AUC: 0.74). The distress score of the mice in all treatment groups indicated no prognostic ability at 6–3 days before the humane endpoint (AUC: 0.50–0.64). The burrowing behavior after 2 and 17 h indicated a high predictive power for all the treatments, 2–3 days before the humane endpoint (AUC: 0.88–1.00). The nesting activity of the mice proved a good discriminatory power up to two days before the humane endpoint (0.84–1.00) throughout the treatments. The MGS score was predictive 2–3 days in advance for the vehicle, LXH-254, and the combinatorial treatments (AUC: 0.80–1.00), while for the erlotinib-treated mice, the predictive power showed a reduction (AUC: 0.75) on day 2 before the humane endpoint (Figure 4A–C).

To evaluate the treatment strategy with BI + Tram or the triple treatment BI + Tram + BKM, the metastasizing animal model with C57BL/6NTac mice was used. These mice displayed continuous body weight reduction towards the humane endpoint. The body weight of the mice in the vehicle group was significantly reduced on the last three days compared to the weight during tumor progression (Figure 5A). An obvious increase in distress was observed in the mice for all the treatment groups two days before the individual humane endpoint. Here, the mice in the vehicle group indicated the highest scores, with significant differences on the day of humane endpoint and one day prior, compared to the tumor progression phase (Figure 5B). The perianal temperature of the mice remained constant, except for two mice in the BI-Tram treatment group at the humane endpoint (Figure 5C). The murine burrowing performance steadily decreased in all therapy groups both after two hours and 17 h (Figure 5D,E). The nesting activity of some mice was reduced towards the humane endpoint, while one mouse in the BI + Tram treatment group was able to build a good nest even at the humane endpoint (Figure 5F).

The ROC curve analysis of the metastasizing PDA model indicated a good predictive power for body weight five days prior to the humane endpoint for the Tram + BI treatment group for B6NTac mice (AUC: 1.00, Figure 6). However, for the vehicle and triple-treated mice, a good diagnostic ability of body weight was quantified earliest two days prior to the humane endpoint (AUC: 0.89–1.00). The distress score of the vehicle and triple-treated mice indicated a good prognostic power two days prior to the humane endpoint (AUC: 0.83–0.94), while the prediction for the humane endpoint in the Tram + BI group was rather low on the days prior to the humane endpoint (AUC: 0.75). The burrowing behavior of the mice after 2 and 17 h indicated a good predictive power two days prior to the humane endpoint in both therapy groups (AUC: 0.89–1.00). For the vehicle-treated mice, a good prediction for the humane endpoint was possible with the burrowing behavior after 2 h at least one day before the humane endpoint. The nesting activity of the mice revealed a low diagnostic ability for both therapy groups (AUC: 0.56–0.78) and a high prediction for the vehicle group (AUC: 0.90–1.00, Figure 6).

The same therapeutic strategy was performed in the metastasizing PDA model with wildlings. The wildlings indicated continued reduction in body weight and an increased distress score towards the humane endpoint (Figure 7A,B). The perianal temperature remained stable, except for a reduction at the humane endpoint in one vehicle and one BI + Tram-treated mouse (Figure 7C). Also, for the behavioral parameters, the burrowing behavior and nesting activity, a continuous decline towards the endpoint could be recognized in the wildlings (Figure 7D–F).

The ROC curve analysis of all the parameters was performed for the wildlings assigned to the vehicle group only (Figure 8). Since both treatment groups included merely one mouse, a clear statement about the predictive power of the parameters for humane endpoint determination is not scientific. For the vehicle-treated wildlings, the body weight indicated a low prognostic ability two days prior to the humane endpoint with an AUC of 0.78. The humane endpoint prediction from the distress score proved to be slightly better, with an AUC of 0.83 five days before the humane endpoint. The predictive ability of the perineal temperature of the mice was rather low at 1–2 days before the humane endpoint (AUC: 0.58–0.67). In contrast, the behavioral parameters of burrowing and nesting indicated a good diagnostic ability 3–2 days prior to the humane endpoint (AUC: 0.83–1.00; Figure 8). When combining the results of all the ROC curves in the present study, the burrowing behavior after 2 and 17 h had an especially high predictive power at least two days prior to the individual humane endpoint for each mouse, independent of the specific PDA model or the therapeutic intervention. On the other hand, parameters such as body weight change, nesting behavior, and the MGS score displayed a high predictability for humane endpoint determination in distinct PDA models and groups, while the distress score and perianal temperature appeared not to be meaningful in the majority of groups.

## 4. Discussion

In the present study, the distress of the mice in different pancreatic cancer models was quantified before any intervention, during the tumor progression, and 1–6 days before the humane endpoints were reached. The predictive power of each parameter to distinguish between animals during tumor progression and animals reaching the humane endpoint was retrospectively analyzed via ROC curve analysis. This statistical approach revealed that the burrowing behavior, both after 2 h and 17 h, indicated a high prognostic ability at the earliest time point out of the analyzed criteria for early humane endpoint determination in all the different pancreatic cancer models, mouse strains, wildlings, and after different interventions.

Burrowing behavior has been commonly used for 10 years to assess postoperative [35] and neuropathic pain [36], the severity in different animal models [27,28], and to compare different pain management protocols [37,38]. In line with the consensus in the literature, our results showed that burrowing behavior is a sensitive parameter for the quantification of welfare impairment or pain in laboratory animals. Burrowing behavior might be an early humane endpoint criterion, which was presented in our previous study, where the reduction in burrowing behavior, together with 10% body weight loss, was able to predict the humane endpoint within two days in a murine bile duct ligation model [23]. Besides burrowing behavior, nesting activity seems to be predictive of early humane endpoint determination in some therapeutic groups. Nesting activity was already used to assess pain-related depression [39], to quantify differences between analgesics [37,38], and also for severity assessment in different animal models [28,40]. Nesting is, therefore, also a sensitive parameter for the assessment of animal welfare. When mice are suffering, they might not be able to build a good nest, potentially leading to hypothermia, which thus can contribute to further deterioration in their condition. In consideration of this aspect, the usefulness of nesting activity as a humane endpoint criterion should be critically questioned. This negative aspect does not account for burrowing behavior, since the burrow can be placed in addition to the conventional nesting material in the home cage.

Body weight change proved to be predictive for a limited number of treatment groups towards the humane endpoint. Paster et al. had already declared that body weight loss alone might not be a good humane endpoint criterion in oncological animal models, since the tumor weight gain can compensate for the loss of tumor-free body weight [41]. This hypothesis is reflected by the fact that no significant loss of body weight was observed for mice after subcutaneous or intravenous injections or for some therapeutic groups, even on the day of the humane endpoint. However, this is not valid for all cancer models, since the humane endpoint could be predicted by body weight loss with a sensitivity of 97% in a glioblastoma model in rats [42]. Additionally, body weight loss proved to be a good humane endpoint criterion for sepsis studies [43] or disseminated sporotrichosis [44]. While 20% body weight loss is used in most studies as a humane endpoint criterion, in some studies for colitis and chronic diabetes, the mice lost more than 20% of their body weight, without any signs of pain or distress, and were able to recover [18]. The early euthanasia of these mice would lead to the unnecessary loss of scientific data and an increase in animals [18]. Body weight alone is, therefore, not a good criterion in some kinds of animal models, but together with other criteria, such as body temperature [45], burrowing [23], or body temperature in combination with clinical scores [24], it proved to be a precise predictor for the humane endpoint in animal models for infection, sepsis, stroke, and bile duct ligation.

The distress score determines the humane endpoint and indicates, therefore, an AUC of 1.00 on the day of the individual endpoint in all the groups in the present study. However, the predictive ability, a few days before the humane endpoint, is rather bad. One could also hypothesize that more sensitive observation criteria need to be defined on the score sheet to detect the impairment of welfare earlier. However, mice are prey animals and usually suppress any signs of pain and discomfort. The first sign of discomfort is piloerection, which is listed on the clinical score. Criteria that are more sensitive might be difficult to define. The low AUC values illustrate that the clinical score alone is not sufficient for early humane endpoint determination and should be replaced by more sensitive parameters, as already described in the literature [46].

The prediction of the humane endpoint via perianal temperature is rather bad in all the PDA models and treatment groups. A decrease in body temperature is known as the most sensitive humane endpoint criterion for mice with infectious diseases, as described for venom snake exposure [47,48], ESKAPEE infection [49], or CLP-induced sepsis [50]. These infectious diseases lead to a rapid impairment of health and hypothermia. While murine cancer models indicate a slower impairment of health, the mice will be euthanized before hypothermia might occur, due to the occurrence of endpoint criteria such as 15–20% body weight loss, bent body posture, or abnormal breathing. The measurement of perianal surface temperature might also not be sensitive enough, compared to the body core temperature.

The MGS indicated a good humane endpoint prediction for a few groups but was not as predictive as burrowing. The MGS is the most important parameter to assess pain in mice [51,52]. Due to the continued administration of metamizole in their drinking water, the pain might be adequately treated, and pain-related behavior is not observable. A clear limitation of this study is that MGS was not scored in the murine metastasizing animal models. The assessment of the MGS by using two different methods, such as video recording and observation in the home cage, might also cause potential bias in the results of the present study.

The humane endpoint criteria for euthanasia differed especially between the different cell-derived animal models and with the Panc02 cells. The intravenously injected mice were euthanized due to abnormal breathing, whereas the subcutaneously injected mice were euthanized because of tumor ulceration. The occurrence of tumor ulceration was not predictable and was often observed in mice, which indicated no signs of discomfort on the day before. The determination of early humane endpoint criteria for subcutaneously injected mice is therefore difficult. However, in comparison to the intravenously and orthotopically injected mice, the subcutaneously injected mice indicated the lowest distress on the day of the humane endpoint. For the orthotopic models, including the different therapeutic groups, the most commonly observed humane endpoint criteria were body weight loss, bent body posture, or ascites. The examination of the dead animals revealed that the stress was caused mostly by the ingrowth of the tumor into the intestine. Unfortunately, some of the used drugs promoted the invasive growth and were not used for further preclinical validation. Other side effects of the drugs were not observed. The orthotopic models with the Panc02-injected cells were euthanized at a body weight loss of 15%, while for the metastasized models, 20% body weight loss was used as the predefined endpoint criterion. Despite the various animal models, endpoint criteria, mouse strains, wildlings, and treatment groups, burrowing proved to be a robust predictor for the humane endpoint in all pancreatic cancer models. The advantage of burrowing is that it can be scored easily and is an objective criterion, while parameters such as nesting, the distress score, and the MGS score are also strongly affected by observer bias. Objective assessment of the humane endpoint is a key factor to ensure reproducibility, especially when different investigators are involved in the same research project.

A clear limitation of burrowing is that mice need to be single housed. Female mice can be separated for the night to monitor burrowing behavior and put back into the group without any problems on the next day. This is difficult with male mice, reuniting them could cause severe rank fights. However, recent studies revealed that single housing of mice is not more stressful compared to group housing [53,54,55].

The combination of burrowing with another predictive criterion could increase the prognostic ability, as was already shown with body weight loss in a bile duct ligation model in one of our previous studies [23]. The combination of other parameters, such as body temperature with body weight or clinical score, proved to be sensitive for humane endpoint assessment [45,56]. Even one machine learning approach has been published, where temperature, body weight change, and data from a neuro score were combined to define humane endpoints in murine models for sepsis and stroke [24]. However, huge data sets from many animals are necessary to perform machine learning. Due to the low number of animals, such an approach was not possible in the present study.

The assessment of many different welfare parameters is time-consuming, requires staff, and causes additional stress for the animals due to handling procedures or disturbances in the home cage during the resting phase. In particular, body weight assessment, measurement of perianal temperature, and the MGS in the box require a short and potentially stressful restraint period for the mice. The assessment of burrowing and nesting activity is based on voluntary behavior in the home cage, without causing additional stress. To minimize the stress on mice, the quantification of the most sensitive welfare parameter for a specific animal model in future studies is important.

The aim of the present study was to identify a robust parameter for the early humane endpoint prediction to enable the single-parameter assessment of animal welfare in future studies. The burrowing behavior of the mice after 17 h indicated a high predictive power (AUC: 0.83–1.00) at least two days before reaching the individual humane endpoint in all the orthotopic pancreatic cancer models. This diagnostic ability was achieved with a reduction in burrowing behavior of 44–99% two days before and 75–97% one day before reaching the individual humane endpoint (Appendix A). Further investigation is necessary to evaluate the potential of burrowing behavior as an early humane endpoint criterion for all kinds of animal models. In addition, the basic burrowing performance in the mice is required. In the present study, we set the threshold in the pre-phase at 50 g of burrowed pellets in 2 h and overnight. Using this threshold, in total, 16% of the mice were excluded from the 2 h burrowing analysis, and 7% from the burrowing behavior overnight. A clear limitation of burrowing behavior is that it might not be the method of choice for 100% of the mice, because some individuals might not burrow, even if they are healthy. Some mouse strains, especially with the C57BL/6 background, indicate good burrowing behavior, while some S129 strains do not burrow [32]. But it is likely that burrowing might be applicable as an early humane endpoint criterion for many different animal models, since burrowing performance is known to reflect distress, pain, and an impairment of welfare in animal models for pancreatitis [37,57], bile duct ligation [23], liver fibrosis [58], osteotomy [38], and depression [59].

## 5. Conclusions

The present study revealed that burrowing behavior is a robust predictor of the humane endpoint two days in advance for different pancreatic cancer models and under different therapies.

## Figures and Tables

**Figure 1 animals-15-01241-f001:**
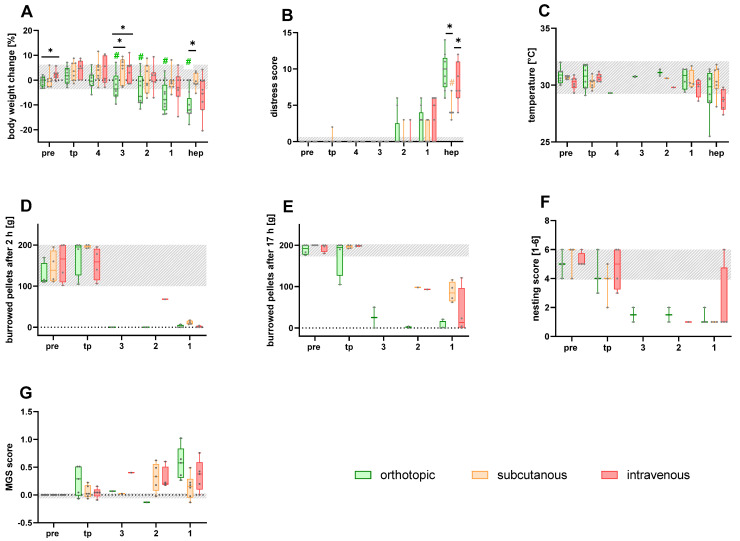
Assessment of the various welfare parameters on healthy mice (pre), during tumor progression (tp), and on the days (4–1) directly before or at the humane endpoint (hep), respectively, in the orthotopic, subcutaneous, and intravenous pancreatic cancer model. The body weight change (**A**), distress score (**B**), and perianal temperature (**C**) were monitored on the four days before and on the day of hep (4, 3, 2, 1, and hep). The number of burrowed pellets after 2 and 17 h (**D**,**E**), nesting activity (**F**), and MGS score (**G**) were assessed for three days prior to the hep. The statistical analysis was carried out with repeated measures two-way ANOVA (**A**,**B**) or a mixed model (**G**), followed by Dunnett’s test for comparisons with tumor progression values (tp) within the models (# *p* ≤ 0.05) and Tukey’s test for inter-model differences (* *p* ≤ 0.05). The grey area represents the baseline values of healthy mice before the tumor cell injection. Orthotopic (*n* = 3–9); subcutaneous (*n* = 3–7); intravenous (*n* = 4–7).

**Figure 2 animals-15-01241-f002:**
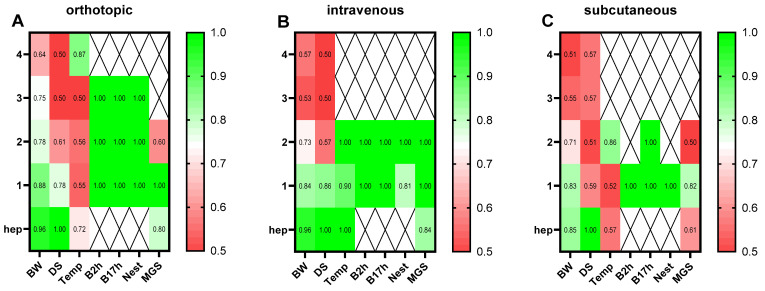
Discriminative power of the welfare parameters for early humane endpoint prediction in different pancreatic cancer models. The predictive power was calculated for each parameter on the days before (4–1) and on the day of the humane endpoint (hep) via receiver operating characteristic curve (ROC) analysis, in comparison to the respective values during tumor progression. The AUC values were calculated, respectively, for the orthotopic (**A**), intravenous (**B**), and subcutaneous (**C**) animal models. The AUC values were displayed via heat maps, indicating the parameters of high predictive power in green (AUC: 0.80–1.00) and low predictive power (AUC: 0.50–0.70) in red. BW: body weight change; DS: distress score; Temp: perianal temperature; B2h: burrowing activity after 2 h; B17h: burrowing activity after 17 h; Nest: nesting activity after 17 h; MGS: mouse grimace scale; orthotopic (*n* = 3–9); intravenous (*n* = 4–7); subcutaneous (*n* = 3–7).

**Figure 3 animals-15-01241-f003:**
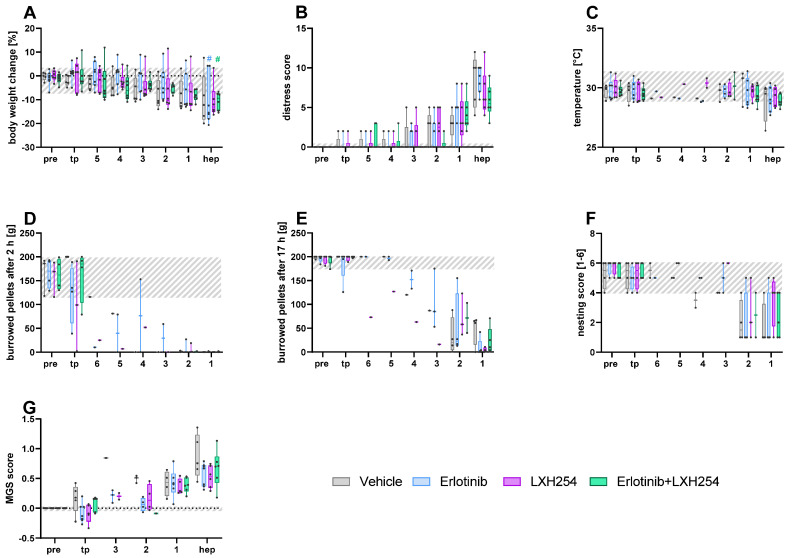
Monitoring of the various welfare parameters on healthy mice (pre), during tumor progression (tp), and on the days (5–1) directly before or at the humane endpoint (hep), respectively, for the different therapy groups in an orthotopic pancreatic cancer model. The percentage of body weight change (**A**), the distress score (**B**), and perianal temperature (**C**) were monitored on the six days before and on the day of the hep (6, 5, 4, 3, 2, 1, and hep). The number of burrowed pellets after two hours (**D**) was analyzed for four days prior to hep and on the day of hep. The number of burrowed pellets after 17 h (**E**), nesting score (**F**), and MGS score (**G**) were analyzed for three days before and on the day of the hep. The statistical analysis was carried out with repeated measures two-way ANOVA (**A**,**B**), followed by Dunnett’s test for comparisons with tumor progression values within the models (# *p* ≤ 0.05) and Tukey’s test for inter-model differences. The grey area represents the baseline values for each parameter of all the mice before the tumor cell injection. Vehicle (*n* = 3–5); Erlotinib (*n* = 4–7); LXH254 (*n* = 3–6); Erlotinib + LXH254 (*n* = 4–7).

**Figure 4 animals-15-01241-f004:**
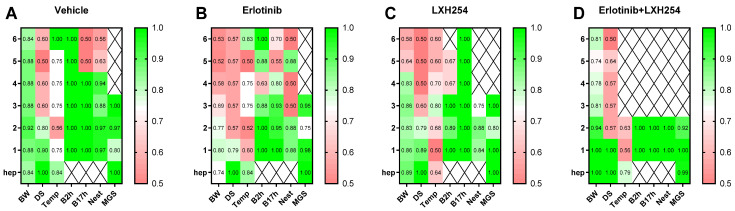
Discriminative power of the welfare parameters for early humane endpoint (ehep) determination for the different therapy groups in an orthotopic pancreatic cancer model. The diagnostic power for ehep prediction was calculated for each parameter on the days before (6–1) and on the day of the humane endpoint (hep) via receiver operating characteristic curve (ROC) analysis, in comparison to the respective values during tumor progression. The AUC values were calculated for the vehicle- (**A**), Erlotinib- (**B**), LXH254- (**C**), and dual Erlotinib/LXH254-treated (**D**) animals. The AUC values were displayed via heat map, indicating parameters of high predictive power in green (AUC: 0.80–1.00) and low predictive power (AUC: 0.50–0.70) in red. BW: body weight development; DS: distress score; Temp: perianal temperature; B2h: burrowed activity after 2 h; B17h: burrowed activity after 17 h; Nest: nesting activity after 17 h; MGS: mouse grimace scale; vehicle (*n* = 3–5); Erlotinib (*n* = 4–7); LXH254 (*n* = 3–6); Erlotinib + LXH254 (*n* = 4–7).

**Figure 5 animals-15-01241-f005:**
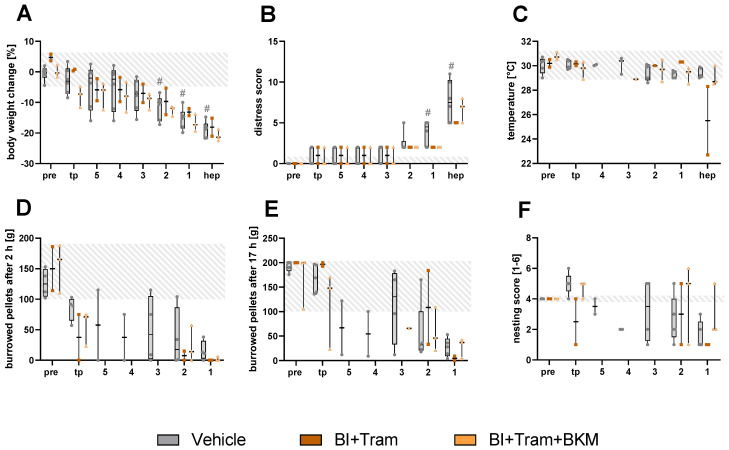
Analysis of the various welfare parameters on healthy C57BL/6NTac mice (pre), during tumor progression (tp), and on the days (5–1) directly before or at the humane endpoint (hep) for the different therapy groups in a metastasizing tumor model. The percentage of body weight change (**A**), distress score (**B**), perianal temperature (**C**), number of burrowed pellets after two and 17 h (**D**,**E**), and nesting score (**F**) were assessed for five days prior to the hep and on the hep (5, 4, 3, 2, 1, and hep). The statistical analysis was carried out with repeated measures two-way ANOVA (**A**,**B**), followed by Dunnett’s test for comparisons with tumor progression values within the models (# *p* ≤ 0.05) and Tukey’s test for inter-model differences. The grey area represents the baseline values for each parameter of all the mice before the tumor cell injection. Vehicle (*n* = 4–6); BI + Tram (*n* = 2); BI + Tram + BKM (*n* = 3).

**Figure 6 animals-15-01241-f006:**
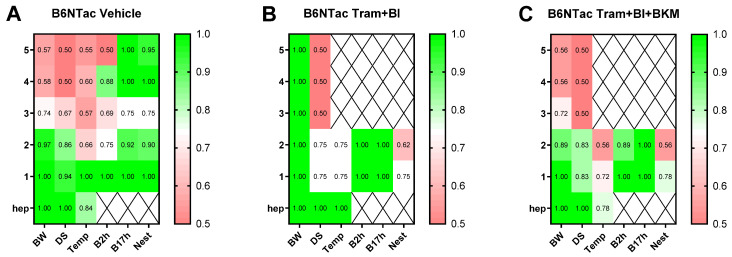
Diagnostic ability of the welfare parameters for early humane endpoint prediction for the therapy groups in a metastasizing pancreatic cancer model. The discriminatory power was calculated on the days before (5–1) or on the day of the humane endpoint (hep) via receiver operating characteristic curve (ROC) analysis, in comparison to the respective tumor progression values. The AUC values were calculated for the different treatment groups, namely vehicle (**A**), dual treatment with trametinib and BI-3406 (**B**), and triple treatment with trametinib, BI-3406, and BKM120 (**C**). The AUC values were displayed via heat maps, indicating the parameters of high predictive power in green (AUC: 0.80–1.00) and low predictive power (AUC: 0.50–0.70) in red. BW: body weight development; DS: distress score; Temp: perianal temperature; B2h: burrowed amount after two hours; B17h: burrowed amount after 17 h; Nest: nesting activity after 17 h; MGS: mouse grimace scale; vehicle (*n* = 4–6); Tram + BI (*n* = 2); Tram + BI + BKM (*n* = 3).

**Figure 7 animals-15-01241-f007:**
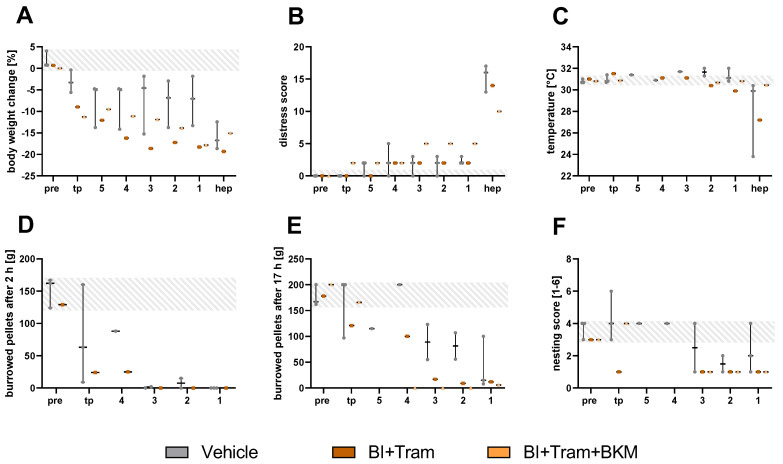
Monitoring of animal welfare using the different parameters on healthy mice (pre), during tumor progression (tp), and on the days (5–1) directly before or at the humane endpoint (hep) for different therapy groups in a metastasizing pancreatic cancer model. The percentage of body weight change (**A**), distress score (**B**), perianal temperature (**C**), number of burrowed pellets after 2 (**D**) and 17 h (**E**), and nesting activity (**F**) were monitored for the four to five days before and on the day of the humane endpoint (5, 4, 3, 2, 1, and hep). The statistical analysis was carried out with repeated measures two-way ANOVA (**A**,**B**) or a mixed-effects model (**D**), followed by Dunnett’s test for comparisons with baseline values within the values with the models and Tukey’s test for inter-model differences. The grey area represents the baseline values for each parameter in the healthy mice before the tumor cell injection. Vehicle (*n* = 3); BI + Tram (*n* = 1); BI + Tram + BKM (*n* = 1).

**Figure 8 animals-15-01241-f008:**
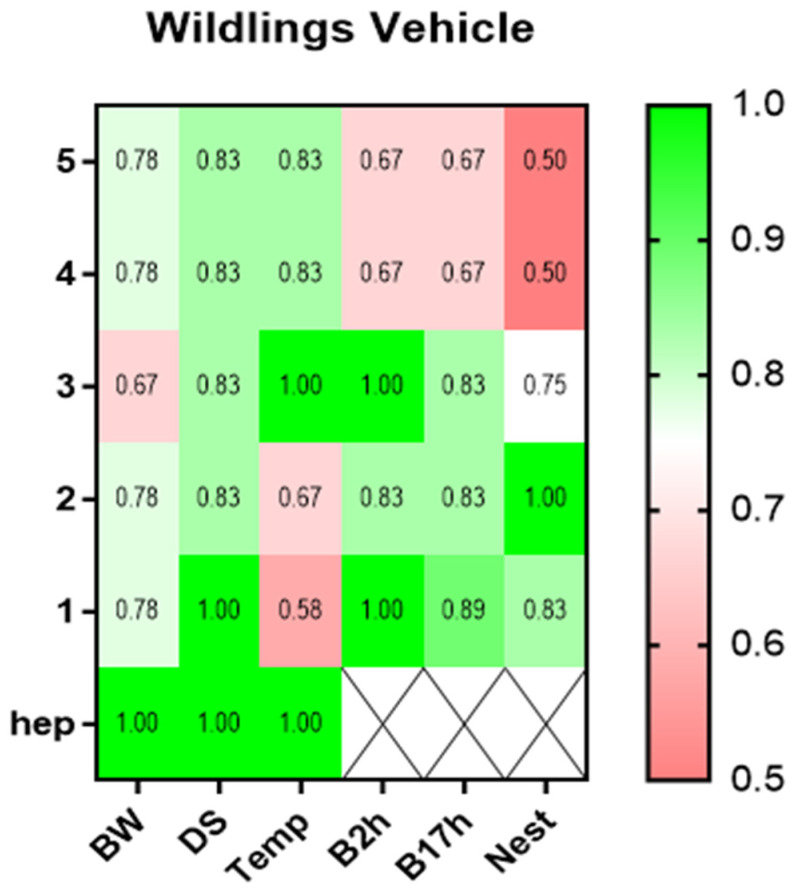
Predictive power of the welfare parameters for early humane endpoint determination for a therapy group in a metastasizing pancreatic cancer model. The discriminatory power was calculated for each parameter on the days before (5–1) or on the day of the humane endpoint (hep) via receiver operating characteristic curve (ROC) analysis, in comparison to the respective tumor progression values. The AUC values were calculated for vehicle-treated animals. The AUC values were displayed via heat maps, indicating the parameters of high predictive power in green (AUC: 0.80–1.00) and low predictive power (AUC: 0.50–0.70) in red. BW: body weight development; DS: distress score; Temp: perianal temperature; B2h: burrowed amount after two hours; B17h: burrowing activity after 17 h; Nest: nesting activity after 17 h; MGS: mouse grimace scale; vehicle (*n* = 3).

## Data Availability

The raw data from the present study are attached in the form of an xlsx. file as Appendix A.

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
