# Peer review of "Burrowing Behavior as Robust Parameter for Early Humane Endpoint Determination in Murine Models for Pancreatic Cancer"

_animals, 2025, doi:10.3390/ani15091241_

Round 1

Reviewer 1 Report

Comments and Suggestions for Authors

Lines 65-70: What about tumor size as endpoint?

Line 105: Please mention approval from the ethical committee

Lines 121 and 127: Please correct "wildlingswere"

Line 190: Please define route of administration 

Line 208-215: It would be better to include the score sheet and the exact criteria for human endpoint.

Line 222: Please verify the dimension of the borrowing tube; 0,03 seems very thin

Tumor progression is used throughout the manuscript, but no time points are given. It would be better to include an overview of the timeline of the models and of the timepoints of the human endpoint for the different animals, perhaps as a supplementary file.

Discussion: Please discuss the implications of the need for single housing when using burrowing as an alternative tool for early detection of human endpoint.

Reviewer 2 Report

Comments and Suggestions for Authors

The manuscript describes the usefulness of several parameters to indicate early humane endpoints in different models for pancreatic cancer in a variety of mouse strains. 

The data have been collected from different experiments. Although this leads to small differences in set-up or outcome, overall this increases the robustness of the outcome. I have some minor remarks for clarification to the reader:

  1. Line 132: why were all animals housed individually? I understand for the sake of several scores, it might be easy, but social housing should be the basic premise.
  2. Line 163-165: I understand that the data were collected from experiments that were performed for other goals. In that sense it is excellent use of data. From an ethical perspective, it would be good to make this more clear in the introduction and abstract. If these severe experiments had been performed for the sole purpose of investigating early HEP, I would find that rather unethical.
  3. L 213-215: What exactly were the criteria that were used for HEP? It seems, for example, that you used BW as one of the HEP criteria. If this is the case, then it is not possible to test BW for early HEP as well, since the data would be very much influenced by the fact that you are using it as HEP criterion. If this is the case for other criteria that you tested as well, than these should be discarded from your analysis.
  4. L 228: You have excluded animals that did not burrow properly. Did you also exclude animals that did not build a proper nest? Nest building is a trait that is very strain dependent.
  5. L233-241: How often did you score the MGS? Daily? If so, that is a very intrusive test for the mice as well as time consuming for the researcher. Also: why did you change from the cage set-up to the home cage set-up? I would presume that this change would drastically affect the outcome of the MGS
  6. Discussion: You have excluded mice that did not burrow properly. You touch upon this briefly in lines 613-615. How many animals did you have to exclude from your test, and how would this affect your conclusion on the robustness of this parameter as humane end point?
  7. Discussion: In general terms, you mention that assessment of welfare parameters is time consuming and causes stress to the animals. Would it be possible to make this a bit more quantifiable in your dataset? These are, in my opinion, important aspects to include to establish the robustness
